# Sources of Dietary Salt in North and South India Estimated from 24 Hour Dietary Recall

**DOI:** 10.3390/nu11020318

**Published:** 2019-02-01

**Authors:** Claire Johnson, Joseph Alvin Santos, Emalie Sparks, Thout Sudhir Raj, Sailesh Mohan, Vandana Garg, Kris Rogers, Pallab K. Maulik, Dorairaj Prabhakaran, Bruce Neal, Jacqui Webster

**Affiliations:** 1The George Institute for Global Health, The University of New South Wales, Sydney, NSW 2006, Australia; jsantos@georgeinstitute.org.au (J.A.S.); esparks@georgeinstitute.org.au (E.S.); krogers@georgeinstitute.org.au (K.R.); bneal@georgeinstitute.org.au (B.N.); jwebster@georgeinstitute.org.au (J.W.); 2The George Institute for Global Health, Hyderabad, Telangana 500034, India; traj@georgeinstitute.org.in (T.S.R.); pmaulik@georgeinstitute.org.in (P.K.M.); 3Public Health Foundation of India, New Delhi, Delhi 110017, India; smohan@phfi.org (S.M.); vgarg@phfi.org (V.G.); dprabhakaran@ccdcindia.org (D.P.); 4Centre for Chronic Disease Control, New Delhi, Delhi 110017, India; 5Charles Perkins Centre, University of Sydney, Sydney, NSW 2006, Australia; 6School of Public Health, Faculty of Medicine, Imperial College London, Kensington, London SW7 2AZ, UK; 7Royal Prince Alfred Hospital, Sydney, NSW 2050, Australia

**Keywords:** salt, sodium, dietary recall, nutrition, public health, cardiovascular disease

## Abstract

Recent data on salt intake levels in India show consumption is around 11 g per day, higher than the World Health Organization’s (WHO) recommended intake of 5 g per day. However, high-quality data on sources of salt in diets to inform a salt reduction strategy are mostly absent. A cross-sectional survey of 1283 participants was undertaken in rural, urban, and slum areas in North (*n* = 526) and South (*n* = 757) India using an age-, area-, and sex-stratified sampling strategy. Data from two 24-h dietary recall surveys were transcribed into a purpose-built nutrient database. Weighted salt intake was estimated from the average of the two recall surveys, and major contributors to salt intake were identified. Added salt contributed the most to total salt intake, with proportions of 87.7% in South India and 83.5% in North India (*p* < 0.001). The main food sources of salt in the south were from meat, poultry, and eggs (6.3%), followed by dairy and dairy products (2.6%), and fish and seafood (1.6%). In the north, the main sources were dairy and dairy products (6.4%), followed by bread and bakery products (3.3%), and fruits and vegetables (2.1%). Salt intake in India is high, and this research confirms it comes mainly from added salt. Urgent action is needed to implement a program to achieve the WHO salt reduction target of a 30% reduction by 2025. The data here suggest the focus needs to be on changing consumer behavior combined with low sodium, salt substitution.

## 1. Introduction

Cardiovascular diseases (CVDs) are the leading cause of death in India, with high blood pressure responsible for almost one quarter of the 2.3 million CVD-related deaths per year [1]. India has committed to achieving the global target of reducing population salt intake by 30% by 2025 to reduce the burden of non-communicable diseases (NCDs) [2,3]. Determining levels of salt intake and major sources of dietary salt are key steps to reaching this goal. Salt intake in India is about 11 g per day, exceeding the WHO’s recommended maximum intake of 5 g per day [4]. Sources of dietary salt are known to vary between countries, with over 75% of dietary salt in high-income countries coming from processed foods [5], while the predominant source of dietary salt in low- and middle-income countries (LMICs) is from homemade foods where salt is added during food preparation [6].

Data on the salt content of Indian foods are sparse, exacerbated by the lack of compliance with nutrition labelling [7] and absence of a comprehensive, validated nutrition composition database to date [8], which would facilitate a more accurate estimation of major dietary sources of salt from dietary surveys. Current evidence suggests that the main source of dietary salt in India is from added salt during cooking [8]. However, India is presently undergoing a rapid epidemiologic, demographic, and nutrition transition, and salt intake from pre-prepared packaged foods may be increasing as a result, as seen in other countries which have undergone a similar transition [9]. The process of the nutrition transition has three stages: (1) consumers move away from traditional staple items and towards a more ‘westernized’ diet, including the increased consumption of wheat in the form of bread, cakes, and cookies; (2) the influences of globalization mean that consumers have more access to a variety of convenience foods, including processed, ready-to-eat, deep-fried food with added preservatives; and (3) in higher socioeconomic populations, consumers are aware of adverse eating habits and try to adapt to a healthy lifestyle [10]. Most of the urban middle-class population in India is reportedly in the second stage of nutrition transition, and the effect is marked by a decline in the consumption of coarse cereals, fruits, and vegetables, and an increased intake of unhealthy fats, dietary salt, and animal foods [11]. The pervasive presence of food and beverage advertisements, particularly through TV and radio, as well as the rapid growth of supermarkets and fast-food outlets is further contributing to the increasingly imbalanced diets of many Indians [9]. This trajectory is unlikely to change given the global trend towards processed, pre-prepared food in LMICs where the rate of increased consumption of processed foods, supplied by transnational food corporations, food manufacturing, food retail, and catering industries, is now the fastest worldwide [12]. In India, the increasing supply, availability, and access to these foods facilitate low prices, prompting the population to increase their purchases [10]. As a result, not only is awareness of these products increased, but changes in taste preferences emerge [13], contributing to increased demand [10].

Further, with increasing rural to urban migration, there is greater availability and demand for processed food [10]. The arrival of chain restaurants and fast food outlets in urban settings alongside continued consumption of traditional Indian cuisine, which uses substantial amounts of salt for food preparation as well as for seasoning at the table, means that the sources of dietary salt are likely to steadily increase in number.

The above factors highlight the importance of understanding how diets are changing. This study aims to capture dietary sources of salt in populations from North and South India in order to provide a rationale for targeted, diet-related, public health strategies.

## 2. Materials and Methods 

### 2.1. Population and Recruitment

Between February and June 2014, a cross-sectional survey was conducted in an age-, area-, and sex-stratified random sample drawn from urban (slum and non-slum) and rural areas of North and South India. These areas are diverse in terms of the sociodemographic variability in India but were not intended to capture the full range of population groups or dietary habits in the country. In the north (Delhi and Haryana), census enumeration blocks (for urban areas) and villages (for rural areas) were sampled at random from within the study area. In the south (Andhra Pradesh), the census enumeration blocks and villages were selected to be broadly representative of those in the state using a purposive process aligned with existing affiliations, and further detail on upper- and middle-income areas was collected. In both regions, a census list including information about the age and sex of all inhabitants was compiled for selected census enumeration blocks and villages, and a random sample of the population, limited to one person per household ≥ 20 years of age, was invited to participate until recruitment numbers in each stratum were filled [14,15].

### 2.2. Overview of Data Collection and Measurements

The methods for data collection and measurements have been published elsewhere [14,15,16]. Briefly, trained field researchers conducted interviewer-administered questionnaires over two visits within one week which included questions relating to demographics, lifestyle behaviors, disease history, medication use, and knowledge, attitudes and behaviors related to salt, followed by a physical examination. 

### 2.3. Dietary Recall Surveys

Two five-pass, 24-h dietary recall questionnaires were administered through face-to-face interviews by field staff on two days of the same week, and responses were recorded on paper. Staff were trained in the administration of dietary recall surveys following the methodology developed by the Agricultural Research Service of the US Department of Agriculture [17]. Discretionary salt intake was estimated by the collection of information about the quantity of salt added during food preparation and prior to consumption. Participants were asked to estimate the amount of salt added using a food model booklet to assist with the reporting of quantities of foods consumed. Interviewer prompts were provided to account for variable recipe ingredients and food preparation techniques. 

### 2.4. Data Analysis

Data from 24-h dietary recall surveys were transcribed into a purpose-built nutrient database comprising the Indian Food Composition Tables (2016) [18], developed by the Indian National Institute of Nutrition, supplemented by nutritional composition data collected from the labels of all packaged food products sold at Indian supermarkets in Delhi and Hyderabad between 2012 and 2014 [7]. 

Data analyses were conducted in STATA 13 for Windows (StataCorp LP, College Station, TX, USA), and alpha was set at 0.05 significance level. Based on the probability of site selection and household size, sampling weights were developed, then weighted to the reported population structure of the respective regions. The svy command in STATA was used to account for the strata and study design clustering, and the Taylor linearization method for variance estimation [19]. Differences in demographic and clinical variables between North and South India were investigated. 

Weighted mean daily salt intake was calculated by averaging the salt intake estimates of the two 24-h dietary recall surveys collected from each participant. Participants with one or no dietary recalls were excluded. Percent sodium contribution of each food category and subcategory were then computed for each participant by dividing the sodium consumed from each category by the total sodium consumed. The weighted mean percent contributions were then computed and compared between North and South India. Within each region, differences in the weighted mean percent salt contribution of the food categories by sex, age, and area were determined using linear regression. North and South India were reported separately due to the known dietary differences between the two regions.

## 3. Results

A total of 2332 participants were selected for the survey: 1291 from Andhra Pradesh and 1041 from Delhi and Haryana; consent was obtained from 1552 participants: 840 from Andhra Pradesh and 712 from Delhi and Haryana. Of those who consented, 1283 people completed two 24-h dietary recalls (82.7%): 757 participants from Andhra Pradesh (90.1%) and 526 from Delhi and Haryana (73.9%). Of these, 49% and 47.5% were female from Andhra Pradesh and of Delhi and Haryana, respectively. The largest proportion of the Andhra Pradesh weighted sample population were from rural areas (66.5%), followed by 27.7% in urban areas, and in Delhi and Haryana, 48.2% were from rural areas and 42.1% from urban areas. The Andhra Pradesh population had a higher proportion reporting a history of high blood pressure (20.2% vs. 9.4%, *p* = 0.042) and chronic kidney disease (2.4% vs. 0.2%, *p* < 0.001), and a lesser proportion reporting a history of high cholesterol (0.9% vs. 3.1%, *p* = 0.028) than Delhi and Haryana. Approximately 40% of both populations were overweight or obese, and almost 25% had high blood pressure, defined as >140/90 mmHg (Table 1). 

### 3.1. Sources of Salt in the Diet

In both regions, the main source of sodium in the diet was added salt: 87.71% in Andhra Pradesh and 83.45% in Delhi and Haryana. The other main food sources of salt in Andhra Pradesh were from meat, poultry, and eggs (6.33%), followed by dairy and dairy products (2.62%), fish and seafood (1.61%), and fruits and vegetables (1.11%). In Delhi and Haryana, the main sources of salt in foods were dairy and dairy products (6.36%), followed by bread and bakery products (3.34%), fruits and vegetables (2.07%) and snack foods (2.05%). Percent sodium contribution of ten out of twelve categories was significantly different between the two regions, though making small contributions overall. (Table 2).

Within regions, after exclusion of added salt, differences in sex, age, area, and education were evident for some food categories (Appendix A). However, these differences are very small in absolute terms given the contribution of added salt to dietary salt intake.

### 3.2. Proportion of Salt Intake from Added Salt While Cooking or at the Table

Within each region, there was no difference in mean percent contribution of added salt to overall salt intake between males and females (both *p* > 0.780), nor between groups defined by age or education level (all *p* > 0.05). In Andhra Pradesh, urban areas consumed a higher proportion of salt from added salt than those in rural areas: 90% compared to 86% of salt intake (*p* = 0.009). In contrast, the inverse was found in Delhi and Haryana, with 86% compared to 81% in rural and urban areas respectively (*p* = 0.019; Table 3 and Appendix A).

Comparing between regions, a higher proportion of salt was consumed as added salt in Andhra Pradesh compared to its counterparts in Delhi and Haryana for males (*p* = 0.006) and females (*p* = 0.001), participants aged 20–39 years (*p* = 0.010) and 40–59 years (*p* = 0.011), urban slum (*p* < 0.001) and urban areas (*p* < 0.001), and tertiary-level-educated participants (*p* < 0.001).

## 4. Discussion

In both regions and across all areas (urban, urban slum, and rural), the largest source of salt intake was from salt added during the cooking process or at the table—greater than 80 percent. Higher salt intake was reported for urban slums versus rural areas in Andhra Pradesh, although this was not the case in Delhi and Haryana. However, the differences in absolute terms are small and likely not significant from a policy-making perspective. Our study did not find that people in urban areas were eating more salt from packaged or restaurant foods, except for snack foods. This is surprising considering the nutrition transition currently occurring in India and the availability of processed foods in urban areas [10,20]. It is highly probable that this change in diet is in fact currently underway, although it appears not to have progressed as far as previously reported [9,10]. Future research is required to monitor dietary changes and determine other key sources of salt in the diet to identify new and changing dietary sources of salt in order to adapt and modify salt reduction interventions. 

Our findings are similar to research undertaken in recent years. A 24-h dietary recall survey of adults in Tamil Nadu in 2012 showed that 80% of food consumed was home-prepared [8]. There was a marked difference between the food sources of salt intake between rural and urban populations in Tamil Nadu. Similarly, in our study, salt intake from cereals and grains and dairy products were higher in urban areas in Andhra Pradesh, and salt intake from animal meat was higher among the rural and urban slum populations surveyed. In contrast, in Delhi and Haryana, salt intake from dairy sources was higher, and intake from fruit and vegetables was lower in rural areas. This is largely due to the differing diets of each state, whereby animal meat is consumed more frequently in the state of Andhra Pradesh compared with Delhi and Haryana where the population consumes a predominantly lacto-ovo vegetarian diet [21].

Other studies of population diet in India also highlight regional diversity [4,8]. Traditional South Indian diets generally consist of rice, roots and tubers, legumes, coconut, and fish [22]. In North India, wheat and maize are more commonly consumed, as well as poultry and dairy products, including buffalo milk, eggs, and ghee [23,24]. Similarly, in our study, the main sources of salt in the diet varied between regions, but salt added during cooking and at the table contributed more than 80% of overall intake in both. This finding supports the results of a 2016 study which found overall consumption of processed foods was still low [8,25]. 

Compared to the previously published estimate of mean salt intake measured from 24-hour urine samples using the same set of participants [14], the mean salt intake estimated using 24-hour food recall was lower in both regions. The 24-hour food recall method relies on respondents’ self-reports and is prone to recall bias, and several studies have shown that it tends to underestimate salt intake compared to the gold-standard 24-hour urine [26,27,28,29]. In this study, the underestimation was about 16% in South India and 40% in North India. This shows the limited capacity of 24-hour dietary recall surveys to accurately estimate salt intake in populations. Nevertheless, it is a useful tool for identifying sources of sodium in diets [30].

Key strengths of this research are the large size of the populations included and the recruitment of participants from regions in North and South India that span urban slum, urban, and rural populations. Sampling weights were developed from the probability of selection of the site and the number in each household, and these were then weighted for the reported population structures of Andhra Pradesh and Delhi and Haryana. This provides some capacity to generalize the findings to areas of India other than those studied, although there will be parts of India for which significant uncertainty about sources of salt intake remains. Good response rates were achieved, and weighting was used to control for differences in age, sex, and area of residence of those sampled compared with the respective populations of the regions. However, weighting may not have fully adjusted for systematic difference in those that did and did not agree to take part. 

Despite the fact that our survey used two five-pass 24-h recalls, data on condiments, such as pickles, were not captured as a major source of salt in either region in our survey. This may be due to participants underreporting or researcher error during the time of the interview. Regardless, it is a missed opportunity to identify consumption of highly salted foods, which could help to inform targeted consumer education interventions. To capture missed food items, such as condiments, in the future, a short, salt-specific, food frequency questionnaire could be used as a food checklist at the end of the 24-h dietary recall, which might help overcome this limitation [31]. 

Understanding dietary patterns, including major sources of dietary salt, among the population, is a crucial step in developing diet-related, public health strategies such as salt reduction. Preliminary findings from this study were presented to a meeting of government members and stakeholders in 2017 where participants considered a series of actions on salt reduction. In view of the fact that more than 80% of salt in the diet comes from added salt, and it is hard to change consumer behavior, it was felt that reducing salt through use of a salt substitute was an important intervention to consider. In China, where the predominant source of dietary salt is also from added salt [32], trials of salt substitutes have been shown to reduce blood pressure. The main challenge now in both countries is how to overcome the additional costs of salt substitutes, which are almost double the price of standard salt, which means subsidies would need to be considered. Other ideas at the stakeholder meeting included initiating a healthy-eating, mass-media campaign, training for street food vendors, enforcing better labelling of processed foods, and encouraging product reformulation [33]. Whilst current evidence suggests salt reduction action should prioritize reducing added salt in home food preparation, the other proposed interventions were also viewed as important to mitigate the impact of changing diets. Additionally, as populations in India experience the effects of a nutrition transition towards pre-prepared and processed foods, ongoing monitoring of dietary sources, estimated population salt intake, and data on the knowledge, attitudes and behaviors on salt intake will be essential to monitor changes and refocus interventions throughout the lifespan of a national action plan for salt reduction. 

Salt intake in India is about 11 g per day, above the WHO’s recommended maximum intake of 5 g per day [4]. Mathematical modelling suggests that without any decrease in dietary salt intake, Indians in the 40 to 69 year old age group will experience an annual rate of approximately 8.3 million new and recurrent myocardial infarctions, 830,000 new and recurrent strokes, and 2 million deaths from either cause on average during each of the next 30 years. This could be averted by reducing salt intake by 3 g per day over a 30-year period [34]. This presents a unique opportunity to save millions of lives a year in India. Government and stakeholders now need to work together to achieve the 30 percent reduction in salt intake by 2025 committed to in the India’s National Action Plan and Monitoring Framework for the Prevention of Non-Communicable Diseases [3]. 

## 5. Conclusions

Salt intake in India is high, and this research confirms it comes mainly from added salt. Urgent action is needed to implement a program to achieve the WHO salt reduction target of a 30% reduction by 2025. The data here suggest the focus needs to be on changing consumer behavior combined with low sodium, salt substitution.

## Figures and Tables

**Table 1 nutrients-11-00318-t001:** Weighted demographic and clinical characteristics of study populations in Andhra Pradesh and Delhi and Haryana.

Characteristics	Andhra Pradesh	Delhi and Haryana	*p*-Value
Gender (% female)	49.0 (22.0 to 76.6)	47.5 (35.4 to 59.9)	0.937
Age, years (mean, 95% CI)	40.2 (32.7 to 47.7)	40.1 (36.8 to 43.3)	0.969
Age group (%, 95% CI)			
20–39 years	57.6 (29.2 to 81.7)	57.1 (45.1 to 68.3)	
40–59 years	28.7 (10.2 to 58.9)	29.8 (20.4 to 41.2)	0.996
60 years and up	13.7 (4.0 to 38.1)	13.2 (8.5 to 19.9)	
Area (%, 95% CI)			
Urban	27.7 (12.1 to 51.7)	42.1 (31.0 to 54.0)	0.347
Urban slum	5.8 (2.3 to 14.0)	9.7 (6.8 to 13.6)	
Rural	66.5 (41.5 to 84.8)	48.2 (35.9 to 60.8)	
Highest level of education (%, 95% CI)			
Primary level	61.4 (48.8 to 72.7)	45.3 (37.3 to 53.6)	
Secondary level	24.7 (16.7 to 35.1)	30.7 (24.5 to 37.7)	0.059
Tertiary level	13.8 (9.2 to 20.4)	24.0 (18.3 to 30.9)	
Employment (%, 95% CI)			
Employed	66.7 (46.8 to 82.0)	45.5 (35.5 to 55.8)	0.113
Unemployed	33.3 (18.0 to 53.2)	54.5 (44.2 to 64.5)	
Household size, number (mean, 95% CI)	4.6 (4.3 to 4.8)	6.5 (6.0 to 7.0)	<0.001
Body mass index, kg/m^2^ (mean, 95% CI)	24.3 (23.5 to 25.1)	24.7 (23.4 to 26.0)	0.629
Overweight or obese (BMI > 25 kg/m^2^) (%, 95% CI)	38.1 (32.3 to 44.2)	40.9 (33.5 to 48.6)	0.594
Systolic blood pressure, mmHg (mean, 95% CI)	126.4 (121.0 to 131.8)	124.5 (121.7 to 127.3)	0.587
Diastolic blood pressure, mmHg (mean, 95% CI)	78.8 (76.4 to 81.3)	79.9 (78.3 to 81.6)	0.522
Measured high blood pressure (%, 95% CI)	24.3 (16.4 to 34.3)	24.6 (19.2 to 31.0)	0.957
History of high blood pressure (%, 95% CI)	20.2 (11.2 to 33.7)	9.4 (6.2 to 14.0)	0.042
History of high cholesterol (%, 95% CI)	0.9 (0.3 to 2.1)	3.1 (1.5 to 6.1)	0.028
History of heart attack (%, 95% CI)	1.7 (0.9 to 2.9)	2.2 (1.1 to 4.3)	0.574
History of stroke (%, 95% CI)	0.5 (0.2 to 1.8)	0.8 (0.3 to 1.9)	0.598
History of diabetes (%, 95% CI)	7.0 (3.7 to 12.6)	5.0 (3.3 to 7.7)	0.431
History of chronic kidney disease (%, 95% CI)	2.4 (0.9 to 6.1)	0.2 (0.0 to 0.6)	<0.001

**Table 2 nutrients-11-00318-t002:** Mean proportion of daily salt intake derived from different foods categories based on dietary surveys.

	Andhra Pradesh	Delhi and Haryana	Difference	*p*-Value
Mean salt intake, g per day (mean, 95% CI)	8.72 (7.62 to 9.81)	5.62 (5.24 to 6.00)	3.10 (1.82 to 4.37)	<0.001
**Mean Percent Contribution (%, 95% CI)**				
Added salt	87.71 (86.46 to 88.95)	83.45 (81.83 to 85.08)	4.25 (2.30 to 6.21)	<0.001
Beverage (alcoholic)	0.02 (0.00 to 0.04)	0.01 (0.00 to 0.01)	0.02 (−0.01 to 0.04)	0.190
Beverage (non-alcoholic)	0.01 (0.00 to 0.01)	0.21 (0.01 to 0.40)	−0.20 (−0.40 to −0.00)	0.049
Bread and bakery products	0.27 (0.05 to 0.48)	3.34 (2.30 to 4.38)	−3.07 (−4.16 to −1.98)	<0.001
Cereal, grains and products	0.28 (0.24 to 0.31)	1.60 (0.67 to 2.54)	−1.33 (−2.27 to −0.38)	0.006
Dairy and dairy products	2.62 (2.31 to 2.93)	6.36 (5.63 to 7.08)	−3.74 (−4.45 to −3.02)	<0.001
Fats and edible oils	0.00 (0.00 to 0.01)	0.23 (0.04 to 0.42)	−0.23 (−0.42 to −0.04)	0.021
Fish and seafood	1.61 (0.80 to 2.41)	0.01 (0.00 to 0.03)	1.60 (0.79 to 2.41)	<0.001
Fruits and vegetables	1.11 (0.90 to 1.32)	2.07 (1.72 to 2.41)	−0.95 (−1.37 to −0.54)	<0.001
Meat, poultry and eggs	6.33 (5.12 to 7.55)	0.66 (0.39 to 0.94)	5.67 (4.41 to 6.93)	<0.001
Snack foods	0.03 (0.00 to 0.08)	2.05 (1.30 to 2.79)	−2.02 (−2.77 to −1.27)	<0.001
Sugar, honey and related products	0.02 (0.02 to 0.02)	0.01 (0.00 to 0.02)	0.01 (−0.00 to 0.02)	0.213

**Table 3 nutrients-11-00318-t003:** Mean percent contribution of added salt to total dietary salt, by subgroup, derived from 24-h dietary recall for Andhra Pradesh and Delhi and Haryana.

Contribution of Added Salt to Total Salt Intake (%, 95% CI)	Andhra Pradesh	Delhi and Haryana	*p*-Value
Sex			
Male	87.21 (85.19 to 89.24)	82.90 (80.59 to 85.21)	0.006
Female	88.22 (87.14 to 89.31)	84.07 (81.89 to 86.25)	0.001
Age group			
20–39 years	88.04 (86.08 to 90.00)	83.50 (81.17 to 85.83)	0.01
40–59 years	87.29 (85.50 to 89.09)	82.68 (79.66 to 85.70)	0.011
60 years and up	87.20 (86.06 to 88.33)	84.99 (83.05 to 86.93)	0.076
Area			
Urban slum	88.97 (87.29 to 90.65)	83.10 (80.88 to 85.32)	<0.001
Urban	90.49 (88.25 to 92.72)	80.60 (77.34 to 83.86)	<0.001
Rural	86.44 (85.45 to 87.43)	86.01 (84.63 to 87.40)	1
Education			
Primary level	87.17 (86.00 to 88.35)	85.04 (83.18 to 86.90)	0.159
Secondary level	88.30 (86.35 to 90.26)	83.45 (79.36 to 87.53)	0.106
Tertiary level	89.01 (86.43 to 91.60)	80.36 (77.72 to 83.00)	<0.001

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
