# Peer review of "Sources of Dietary Salt in North and South India Estimated from 24 Hour Dietary Recall"

_nutrients, 2019, doi:10.3390/nu11020318_

Round 1

Reviewer 1 Report

The authors contribute important data on salt intake in Indian population subgroups. The manuscript is concise and well written. Comments:

Abstract: Move World Health Organization (WHO) to first use on line 22 and use WHO on line 33.

Introduction:

Line 45 "high income countries COMING from processed foods"

Line 48: "Data on the salt content of Indian foods ARE"

Materials and Methods:

Line 92: "1 person per household ≥20 years OF AGE"

2.3 Dietary Recall

Were recalls administered via computer or paper?

Results:

Table 1.

Specify "Highest level of education"; Define SBP and DBP in footnote.

Absolute salt intake and dietary salt density (mg per 1000 kcals) for each subgroup should be reported in a table. See Morbidity and Mortality Weekly Report, Jan 8, 2016;64(52);1393-7 "Prevalence of excess sodium intake in the US..."

Table 2:

To order the list from highest to lowest contributors seems more logical than listing in alphabetical order.

Discussion: "Contrastingly" change to IN CONTRAST,

Author Response

Thank you for your review. Please find our response attached.

Reviewer 2 Report

General Comment:  The study involves a comprehensive investigation into the sources of salt intake in the northern and southern regions of India.  The authors have compiled an impressive amount of data upon which recommendations can be formed to help reduce salt intake in India. The paper is well written and clear in all areas.  Based on the estimated absolute level of daily salt intake among the population, reduction in dietary salt is urgent.   I agree with the authors that knowing dietary habits in different regions of the country is key to making specific recommendations on how to reduce salt intake. I find the goal of reducing salt intake by 30% in the next five-to-seven years to be quite ambitious.

I do have some issues that I think should be raised.

1.    What were the total salt intakes in both regions?  I failed to see these values in the text or tables, which gave the mean proportion or mean percent of salt intake from different sources.   Did the participants in the study consume the daily average of salt for the population of India, (about11 g/day, line 43)?  

2.    That amount of daily salt intake, 11 g/day, is quite high and indeed constitutes a “high-salt” diet.  It is somewhat surprising to me that with that level of salt intake, about 25% (line 142) of both populations reported with high blood pressure defined as 140/90. (I would expect at least 1/3 of the population with hypertension.) Would these results suggest a low percentage of the population is “salt sensitive”?  Are there other factors, e.g., genetic or environmental, that prevents higher incidence of hypertension in India?  (Just to note, the American Heart Association recently revised blood pressure guidelines to lower what was considered hypertension from 140/90 to 130/80.  If these revised guidelines are valid, the potential for serious cardiovascular events is even more alarming that described in the last paragraph of the paper.)

3.    Is it surprising that with the higher history of high blood pressure in the Andhra Pradesh region that there were no differences in heart attack and stroke histories (Table 1)? Chronic kidney disease was indeed higher and this indicates the strong correlation between hypertension and CKD. 

Author Response

(The authors gave the same response as above.)

Reviewer 3 Report

This large cross-sectional survey in India described that the largest source of salt intake was from salt added during the cooking process or at the table – greater than 80 percent using the data from 24-h dietary recall and also found that people in urban areas were not always eating more salt from packaged or restaurant foods. These epidemiological findings are important for planning strategy for salt reduction programs in India.

I have some comments for consideration

Estimated amounts of salt intake (g/day) were not described. Are salt intake in participants in Andhra Pradesh higher than that in participants in Delhi and Haryana because history of high blood pressure is much higher in participants in Andhra Pradesh? They are the ‘must’ information.

Do some of the participants take antihypertensive medications? If so, it might affect the results of blood pressure.

Last line, please spell out NCD.

In my opinion, estimation of salt intake using spot urine method is easier and cost-effective for this kind of national survey. This can, at least, provide supplementary information regarding salt consumption in India. It could be used for the future study.

Author Response

(The authors gave the same response as above.)

Round 2

Reviewer 3 Report

The authors well responded to my concerns.

One correction, Page 6, line 203, reference [13] should be [14].

Author Response

Thank you very much for reviewing our manuscript and identifying this error in line 203. The reference has now been amended from [13] to [14].